# The Influence of Drying Conditions of Clay-Based Polymer Coatings on Coated Paper Properties

Petronela Nechita 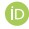

Department of Environmental, Applied Engineering and Agriculture, Engineering and Agronomy Faculty, "Dunarea de Jos" University of Galati, 817112 Braila, Romania; petronela.nechita@ugal.ro

**Abstract:** Coatings based on pigment and polymer binders are applied on paper surfaces to improve their surface, optical, and printing properties. Besides the coating composition, the structure and properties of the coated papers are influenced by the coating layer consolidation upon drying. In this study, the influence of drying conditions on the structure and properties of coating layers based on natural pigments (clay) and polymer binders (butadiene acrylonitrile latex) has been analyzed. Using a laboratory rod Mayer device, the coatings were applied as thin layer (about 15–16 g/m$^2$) on the paper surface and samples of coated paper were dried at 20 and 105 °C temperatures. The optical, structural, and water absorption properties of the coating layer were evaluated by the measurement of gloss, opacity, void fraction, light scattering, and contact angle. The obtained results highlighted that both the drying temperature and latex content in the coating color have a synergic effect on the coated paper quality.

**Keywords:** coatings; coated paper; drying temperature; structure; contact angle; latex



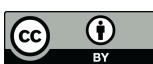

## 1. Introduction

Generally, a coated paper is a layered composite consisting of a paper base covered on both sides with a porous clay-based polymer coating. The coating is applied on the paper surface to improve the gloss, smoothness, optical properties, and printability of the coated paper. The drying of the coated paper is an important stage to obtain a coating layer with a consolidated and appropriate structure and has a great influence on the properties of the coated surface (i.e., gloss, smoothness, printing density, or surface strength). However, besides the coating composition, the type and binder level [1], the pigment shape and size distribution, the properties of the base paper [2], and the coating structure and properties are influenced by the consolidation upon drying [3].

The pigment packing at different binder levels has great influence on the coating layer consolidation during the drying process. When the level of binder is below the critical content for the system, the packing of the pigment (clay) is very disordered. When the level of latex is higher, the consolidation is dominated by the coalescence and the flow of binder upon drying, and the structure becomes more orderly [4]. When the coating formulation is applied onto the base paper, water starts to evaporate and/or penetrate into the base paper. This induces the component pigments and binders to form a dried coating structure. During the drying process, as water evaporates from the coating layer the volume of the remaining coating decreases and surface tensions start to appear, generating capillary force and inducing coating shrinkage [5]. The coating shrinkage often causes poor results in terms of the coated paper properties, such as gloss, light scattering, surface strength, printability, and coating layer uniformity. Recent studies [6,7] have pointed out that understanding the factors associated with coating shrinkage is critical for improving the quality of coated paper products. The main factors which affect the stress during shrinkage are the pigment particle size, the coating layer weight, and the glass transition temperature of lattices.

The shrinkage occurring during the drying of the coating is mainly due to capillary forces as the water recedes in the porous structure [8]. Latex with a high glass transition temperature (*T*g) produces greater drying stress than that with a low *T*g latex. This is attributed to the deformability of the low-*T*g latex. In this context, the drying temperature plays an important role in the development of drying stress. Generally, the drying stress decreases with the increase in the drying temperature because at high temperatures the relaxation of polymer chains occurs [9–11].

When the drying temperature is lower than the *T*g of the latex, cracks develop on the drying surface. In contrast, when the drying temperature is higher than the *T*g of the latex (i.e., 80 °C), surface cracks disappear and leave a uniform film with a few minor cracks and softer outlines. All of the cracks will disappear if the drying temperature is high enough. This is due to the appearance of polymer chain interdiffusion within the coating film [12]. Additionally, at a drying temperature higher than the latex *T*g, the latex particles are soft and under the capillary forces which occur during the dewatering process they are deformed. It is possible to limit the extent of coating shrinkage by using latex with hard particles and a *T*g higher than the drying temperature which does not deform during dewatering [1]. To reduce the influence of coalescence of latex particles on the binding strengths of the coating a post treatment exposing the consolidated coating layer to high temperature can be performed. Although the binding strengths and gloss of coating layer can be improved, this treatment is very limited. An appropriate alternative to solve this drawback seems to be the use of a composite latex formula based on soft and hard polymer. In this case, the soft polymer ensures adhesion and binding strength and hard polymer phase limits the extent of latex particle deformation [12].

In the consolidation of coating layer structure during drying process, two stages are important:

(a) Water evaporation and coating layer consolidation—when the coating is transformed into porous layer which can withstand certain stresses and first critical concentration (FCC) is achieved (Figure 1). The first critical concentration is defined as the solids level where surface gloss begins to decrease rapidly. This is the point where the water excess has been removed from the coating to break the suspension; the pigment particles have formed a semirigid network, which, through restricted consolidation, supports capillary conduction to the evaporation surface.

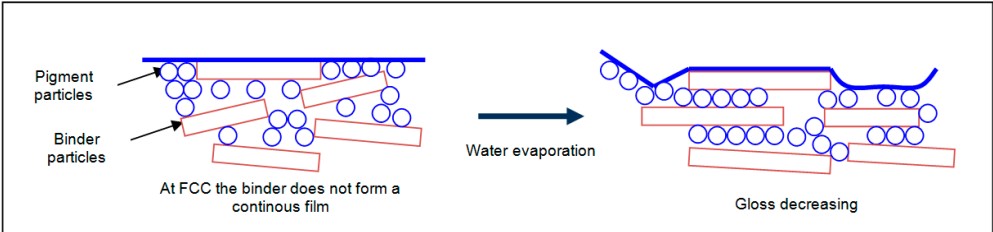

**Figure 1.** The evolution of coating layer properties during the drying process.

(b) Reducing of pores number and latex particles deformation—that occurs shortly after the first critical concentration (this is the stage of second critical concentration). As the water is evaporated the liquid menisci within the pores decreases, the coating layer is contracted and the latex particles are deformed by capillary force; at the end of this stage the water has completely evaporated (the coating layer reached at equilibrium humidity) while the latex particles are not completely deformed (Figure 2) [13–15]. This is the point where the opacity of the coating film begins to increase rapidly. At the second critical concentration, consolidation stops and the capillaries can no longer shrink to accommodate capillary flow to the coating surface. As a result, the capillaries begin to drain the largest reservoirs (fluid filled spaces between pigment particles) to supply the surface with liquid for evaporation [16,17]. The creation of air/solid interfaces causes an increase in the amount of scattered light, thus increasing opacity.

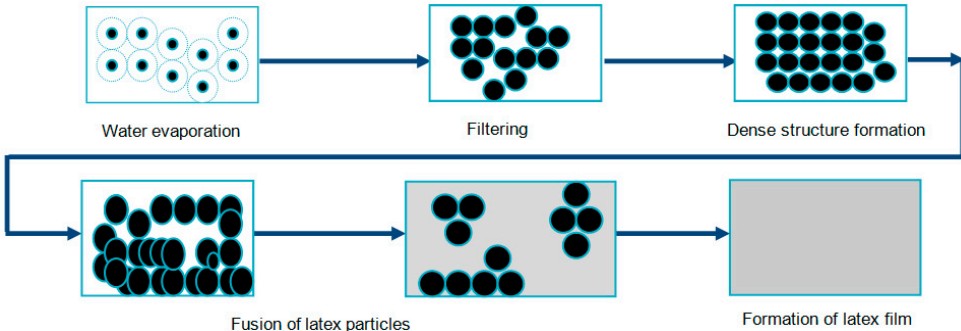

**Figure 2.** The consolidation of the coating layer structure during the drying process.

The paper printability is influenced by coating surface properties, mainly by pores size and their distribution, which have an important role on the paper- ink interactions. The coating binder has a high impact on printability [18–20]. This can be correlated with the drying rate which controls the coating layer dewatering and their structure. During drying process the migration of fluid phase and binder from coating into the base paper and at coating layer surface occurs [21]. This affects the quality of coated paper by appearance of areas with different printing density (mottling phenomenon) [22–24]. In case of high rate drying combined with high absorption of base paper, at contact with coating formulation, the liquid phase and binder penetrate into base paper structure and as result at coated paper surface appear the areas with low binder content resulting a coated surface with high non-uniformity which affects the coated paper printability (Figure 3).

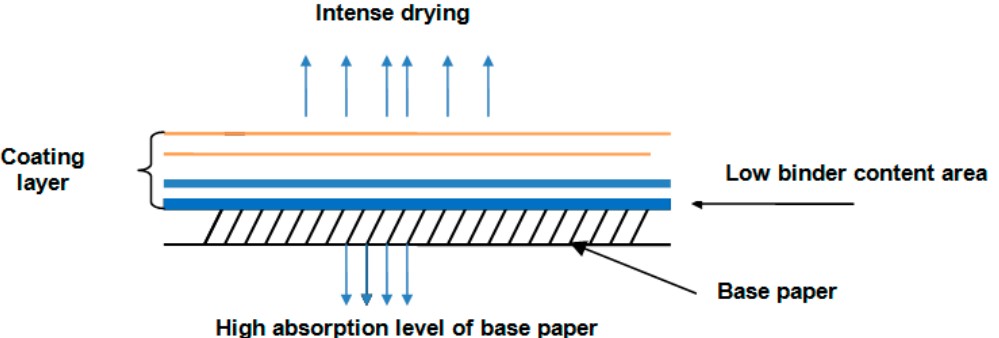

**Figure 3.** Fluid phase migration at a high drying rate.

In this work, the influence of drying conditions on the structure and properties of the coating layer based on natural pigments (clay) and polymer binder (butadiene acrylonitrile latex) has been analyzed. The coating layer was applied in a thin film (15–16 g/m$^2$) on the paper surface and samples of coated paper were dried at different temperatures. The optical, structural, and water absorption properties of the coating layer were evaluated, and based on the obtained results a correlation between the drying temperature and latex content in the coating formulation has been identified.

## 2. Materials and Methods

As a base paper we used a commercial paper (mass and surface sized) with a fibrous composition of 70% bleached hardwood pulp and 30% bleached softwood pulp, with the main properties presented in Table 1.

**Table 1.** The properties of the base paper.

| Properties | Values |
|---|---|
| Grammage, g/m$^2$ | 54.8 |
| Density, g/cm$^3$ | 0.78 |
| Water absorption, Cobb$_{60}$, g/m$^2$ | 22.6 |
| Smoothness, Bekk, s | 50 |
| Breaking length, m | 3703 |
| Brightness, R457/D65, % | 91.43 |
| Opacity, % | 83.44 |
| Gurley porosity, s | 4.5 |

A commercial coating clay (SPS type) with about 74% by weight of particles finer than 2 μm was used as a pigment and a colloidal dispersion of carboxylate butadiene-acrylonitrile copolymer with a glass transition temperature $Tg$ = −20 °C (Brookfield viscosity = 37 mPa·s, particles size of 0.18 μm) as binder.

The composition of the coating formulations is presented in the Table 2 and this was prepared as follows: the pigment was pre-dispersed at 66% dry solids (by weight) in distilled water and, to minimize the dispersion viscosity, 0.3 parts of dispersing agent per hundred parts of pigment (pph) (sodium polyacrylate) was added. The pigment dispersion was diluted to a 45% solid content and different quantities of binder from 5 to 20 pph (parts per hundred parts of dried weight pigment) were added. Sodium hydroxide was used to adjust the pH of the suspension to a value of 8.5 (Table 2).

**Table 2.** The composition and characteristics of the coating formulations.

| Components | Samples Codifications | | |
|---|---|---|---|
| | U1 (20 °C) | U2 (105 °C) | U3 (20 °C + 105 °C 15 min) |
| SPS Clay, pph | 100 | 100 | 100 |
| XNBR Latex, pph | 5 | 10 | 20 |
| Dry content, % | 45.16 | 44.98 | 45.1 |
| Brookfield viscosity, (20 °C, 60 rpm), cP | 120 | 225 | 260 |

A laboratory device based on the Mayer wire-wound rod system was used to apply the coatings in a thin and controlled layer on the base paper surface (about 15–16 g/m$^2$) (Figure 4). The thickness of the coating layer was controlled by the diameter of the wire. A rod 3 value was used, which corresponds to a dry coating layer thickness of about 10 μm.

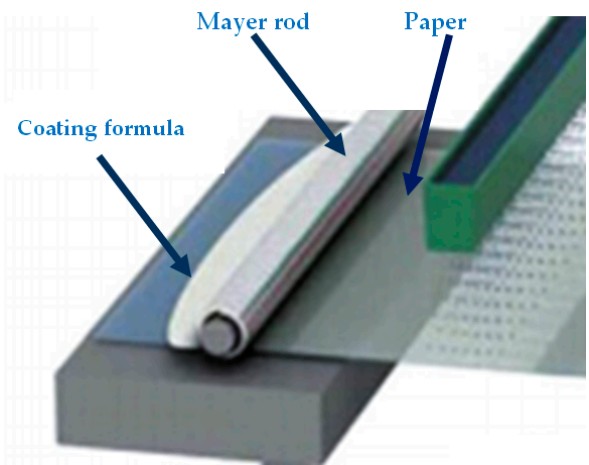

**Figure 4.** Applying the coatings on the paper base surface.

Different drying conditions were used for the coated paper samples: drying at room temperature (20 °C, 6 h)—code sample (U1); in the laboratory oven at 105 °C for 45 min—code sample (U2); in air (at 20 °C, 6 h) and laboratory oven (105 °C, 15 min)—code sample (U3).

The samples of coated paper were analyzed regarding the functional properties, as is mentioned below.

The grammage of the coating layer (g/m$^2$) was measured by the difference between the coated paper and the base paper weight after drying.

To control the coating layer void fraction ($\varepsilon$), the technique of oil absorption was used [20,21]. This is a measure of the pore volume within the coating layer, and the method consists of applying 0.5 mL of oil to a known weight of sample with an area of 30 cm$^2$. After 10 min, when the coating layer becomes translucent, the excess oil has been removed. The sample was weighed again to determine the volume of the oil that was absorbed in the pores. The void fraction, $\varepsilon$, was then determined to be the ratio of the volume of air, $V_a$ (determined from the weight of the absorbed oil and its density) to the total volume of the coating layer, $V_c$ (1).

$$\varepsilon = V_a / (V_a + V_c) \tag{1}$$

where:

$$V_a = W_{oil} / (\varrho_{oil} \times A) \tag{2}$$

and

$$V_c = (f_p \times W_c) / \varrho_p + (f_L \times W_c) / \varrho_L \tag{3}$$

where:

$W_{oil}$—the oil weight;
$\varrho_{oil}$—density of the silicon oil;
$A$—the paper sample surface;
p, L—pigment and latex;
$f$—weight fraction of the component in the coating formula;
$Wc$—the weight of coating layer per unit area;
$\varrho$—the density of each component.

For samples tested on paper, Equation (1) was modified as:

$$\varepsilon = (V_a - V_p) / (V_a + V_c - V_p) \tag{4}$$

where:

$V_p$—the volume of the oil absorbed in the base paper (calculated) with the Equation:

$$V_p = (f_{fibres} \times W_{fibres}) / \varrho_{fibres} + (f_{filler} \times W_{filler}) / \varrho_{filler}) \tag{5}$$

where,

$f$—fraction of fibers and filler in the base paper;
$W$—the weight of fibres and filler per unit area;
$\varrho$—the density of fibres and filler;
$V_a$—the volume of air in the coating layer and the base paper.

Therefore, by subtracting $V_p$ the void volume of the paper is corrected [25,26].

Another parameter used to evaluate the structural properties of the coating layer was the Light Scattering Coefficient (m$^2$/Kg) (LSC), which was measured with an automatic L&W Elrepho SE 070 spectrophotometer (Lorentzen&Wettre, Malmo, Sweden). The method consists of measurements of reflectance at a 457 nm wavelength.

To assess the coating layer wetting capacity, the dynamic contact angle was measured using an automatic device (FibroDat 500/1100, Messmer Buchel, Fokerstrat, The Netherlands) according to the Tappi T558 pm method [27]. The method is based on an image analysis of a water drop (with a specified volume of about 6 μL) in contact with the sub-

strate of a coated paper samples at specified time intervals. The rate of contact angle changes, and the volume and base diameter (spreading on the sample surface) of the water droplet are calculated as time functions.

Image analysis by scanning electron microscopy (SEM) was performed to evaluate the modifications of coating layers in different drying conditions. SEM images were acquired and processed based on the secondary electron stream with a penetration depth of 1–10 μm using the SigmaScan Pro equipment from Systat Software Inc. (San Jose, CA, USA, software version 4.0)

The gloss is the ability of a coating layer to reflect specular light and is a partial measure of the surface quality and shiny appearance of coated paper. The gloss of the obtained coated paper samples was measured according to the SR 7775: 91 [28] method at 75° using a Lehmann device (Lorentzen&Wettre, Malmo, Sweden).

The coated paper opacity (%) was measured using a L&W Elrepho SE 070 spectropho-tometer according to the SR ISO 2471:2001 method [29].

## 3. Results and Discussion

### 3.1. The Influence of Drying Conditions on the Optical Properties of Coating Layer

The latex particle distribution within the coating layer is highly influenced by the drying parameters, and this affects the opacity and gloss of the coating layer. As is presented in Figure 5, at the same drying parameters and with a high latex content in the coating formulation the gloss of the coated surface decreases as a result of latex shrinkage.

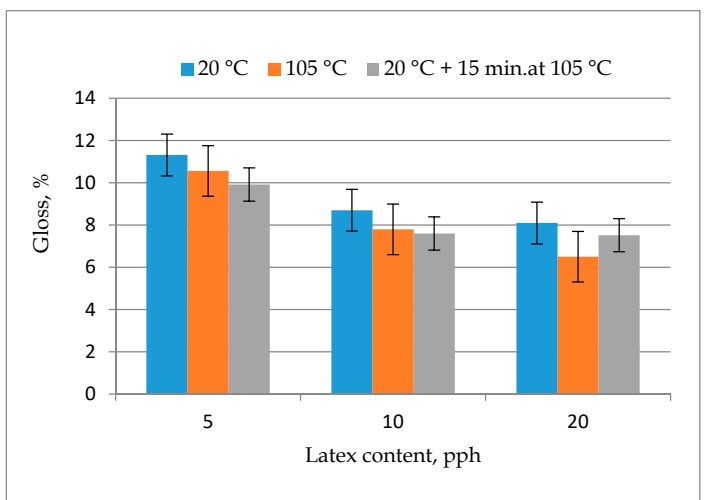

**Figure 5.** The evolution of the coating layer gloss in different drying conditions and with different latex contents.

At a high drying temperature (105 °C) and latex content (20 pph), the latex "melting" phenomenon is intensified and the shrinkage of the coating layer causing the appearance of destruction on the coated surface, which decreases the surface gloss (Figure 6).

Regarding the coating layer opacity, one can observed a different evolution of the coated sample properties depending on the latex content. However, at a lower latex content the coating layer consolidation is influenced by the pigment particle packing, while at a higher latex content the coating layer consolidation is controlled by latex particle coales-cence when the drying temperature is increased. The local stresses which are developed during the coalescence phenomenon determine the particle reordering and accelerate the structural contraction of the coating layer. This leads to a decrease in the coating layer opacity which seems to be more influenced by the latex content than the drying conditions (Figure 7).

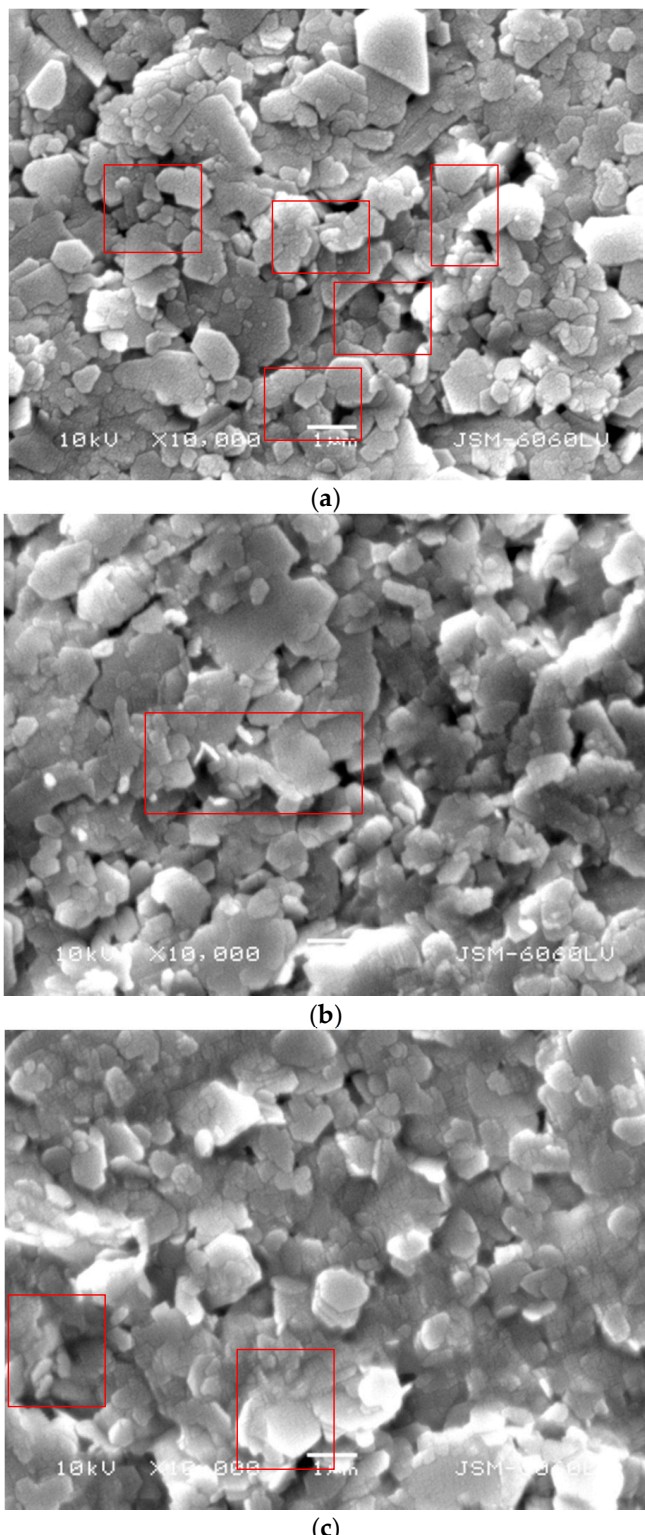

**Figure 6.** SEM micrographs of samples of coated paper with a 20 pph latex content dried in different conditions. The cracks in the coating layer are marked with a red line. (**a**) Drying at 20 °C; (**b**) Drying at 20 °C + 15 min at 105 °C; (**c**) Drying at 105 °C.

*3.2. The Influence of Drying Conditions on the Structural Properties of Coating Layer*

The evolution of the void fraction volume and light scattering coefficient of coated papers dried in specified conditions are presented in Figures 8 and 9.

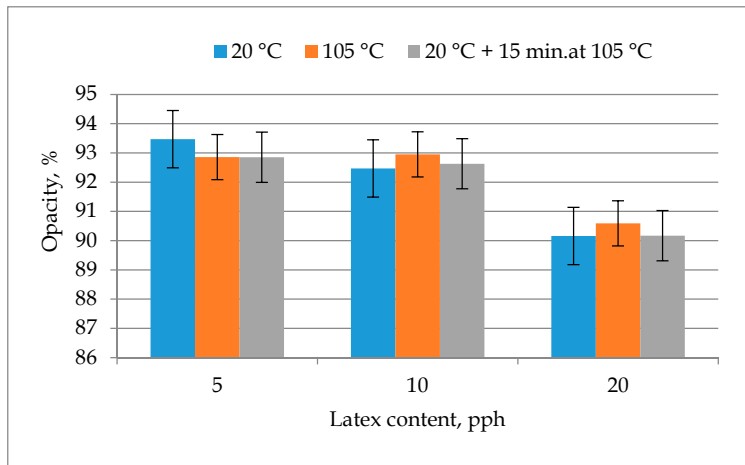

**Figure 7.** The evolution of coated paper opacity at different drying conditions and latex contents.

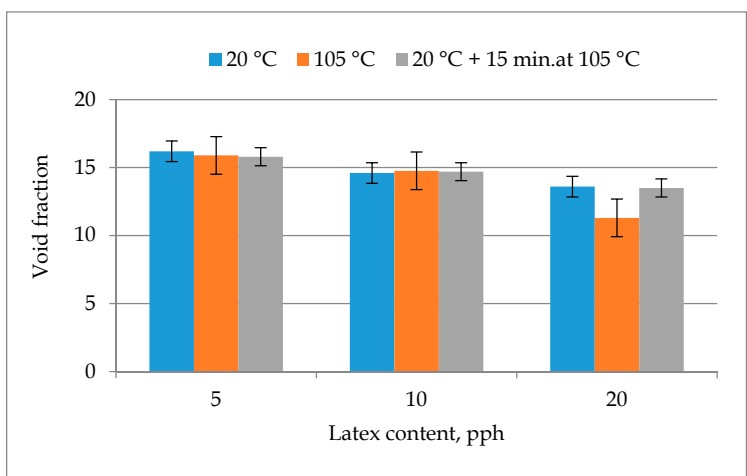

**Figure 8.** The evolution of the void fraction of the coating layer with different drying parameters and latex contents.

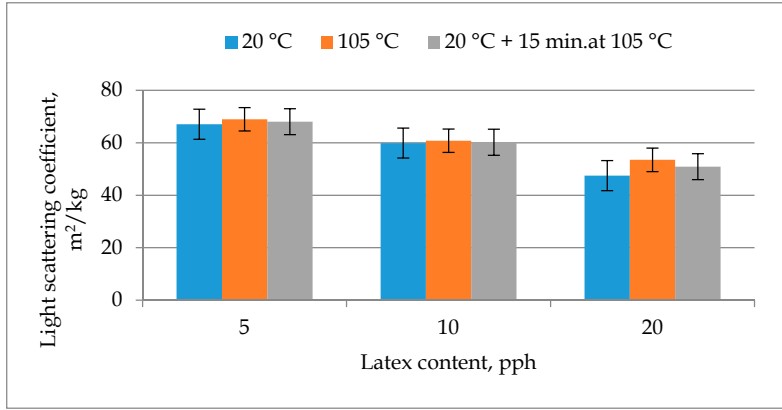

**Figure 9.** The evolution of the light scattering coefficient of the coating layer with different drying parameters and different latex contents.

Analyzing the plotted results, it can be concluded that the structural parameters of the coating layer are influenced by both the drying conditions and the latex content in the coating formulation. At a latex content of between 5 and 10 pph, the drying conditions have a lower influence on the void fraction. As the latex content in the coating formulation

is increased to 20 pph, the intensifying of the drying temperature (105 °C) has the result of obtaining a dense structure of the coating layer with a lower void fraction level. At this latex content, the post treatment of samples by drying for 15 min at 105 °C after drying at 20 °C does not significantly change the coating layer void fraction (Figure 8).

Regarding the light scattering (Figure 9), it can be observed that, at the same latex content in the coating formulation, this parameter is intensified as the drying temperature is increased. The effect is more important at higher latex contents in the coating formulation, although the void fraction is reduced. This can be explained as follows: at high drying temperatures, during latex film formation the particles deform and merge with each other more and more as the latex content in the coating formulation becomes higher. In this case, the pores size in the coating layer structure increases even if the total volume of pores is lower. The change in the pore size when the latex is fully "melted" has the result of the increasing the light scattering for the samples dried at 105 °C and the samples with post treatment drying for 15 min at 105 °C after drying at 20 °C.

This behavior is confirmed by SEM analysis, as presented in Figure 6, where one can observed a dense structure of the coating layer for the samples dried at 105 °C compared with those dried at 20 °C with the constant latex content.

### 3.3. The Influence of Drying Conditions on the Water Absorption and Coating Layer Wettability

The water penetration capacity of the coated papers was evaluated by contact angle measurement. Generally, this parameter provides information about the capacity of fluids to adhere and wet the surface of different substrates. Additionally, the contact angle gives information regarding the substrate's capacity to absorb water-based printing inks. The wetting of coated paper is a complex procedure involving the spreading and absorption of water into the coating layer structure. Absorption starts only after the drops have wetted the surface to a certain extent.

The results concerning the evolution of the contact angle with the drying conditions and latex content of the coated papers are presented in Figure 10. It can be seen that at a latex content of between 5 and 10 pph and an increase in the drying temperature (105 °C), a high water absorption capacity of the coating layer is developed and lower levels of contact angle are obtained. This is the result of latex migration from the coating layer into the base paper, which is higher in coating formulations with a low solids content (aprox. 45%) and a high drying temperature. With latex migration into the base paper, the hydrophobicity level and contact angle value of the coating layer are reduced. At higher latex contents (20 pph) and drying temperature values (105 °C), besides for latex migration contracting occurs with a direct influence on the increase in the coating layer pore size and roughness. As has been mentioned above, the high drying temperature allows the latex to melt and the appearance of large-sized pores, which facilitates the quick penetration of fluids into the coating layer. These phenomena have the result of reducing the contact angle for coating layers with a high latex content that are dried at 105 °C.

It can be observed that a low level of coated surface hydrophobicity is obtained for all the coating formulations. The introduction of a water retention additive (i.e., carboxymethyl cellulose) into the coating formulations and an increase in their solids content (about 60%) seem to be efficient routes to reduce the coating dewatering rate and obtaining coated surfaces with uniform characteristics and an appropriate hydrophobicity [30].

These results are correlated with the gloss values of coating layers with a high latex content and that were dried at 105 °C, the temperature presented in Figure 5.

Figure 11 shows a significant increasing of the drop base diameter, as soon as the drop came into contact with the coated paper surface. After 0.3 s, the rate of change of both contact angle and drop base diameter is low because the surface of coated paper became saturated [31,32]. Decreasing of contact angle of coated surface can be influenced by surface free energy, also. When the paper surface free energy increases, a low level of contact angle is obtained [33].

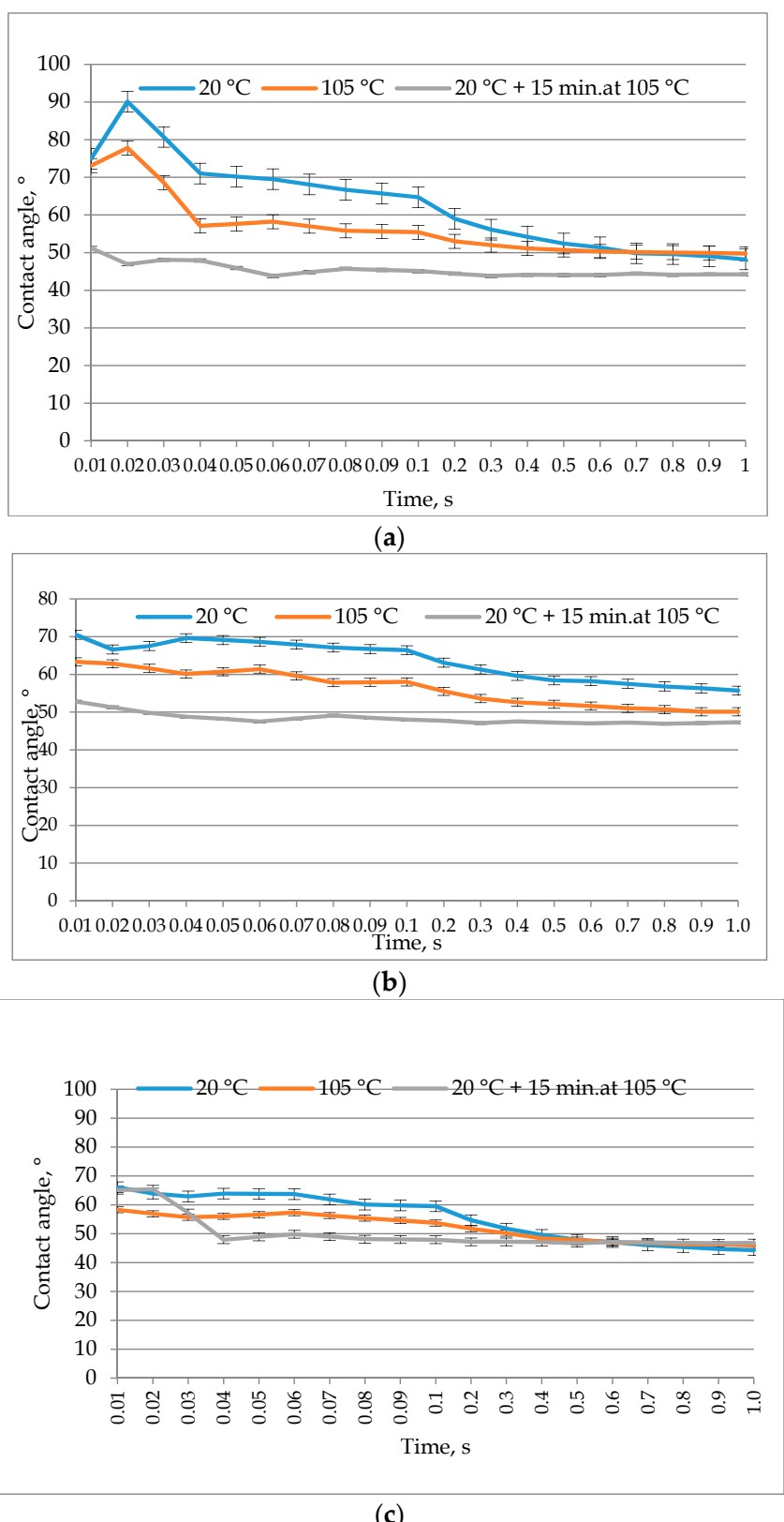

**Figure 10.** The influence of drying conditions on the contact angle of the coated paper samples: (**a**) 5 pph latex content; (**b**) 10 pph latex content; (**c**) 20 pph latex content.

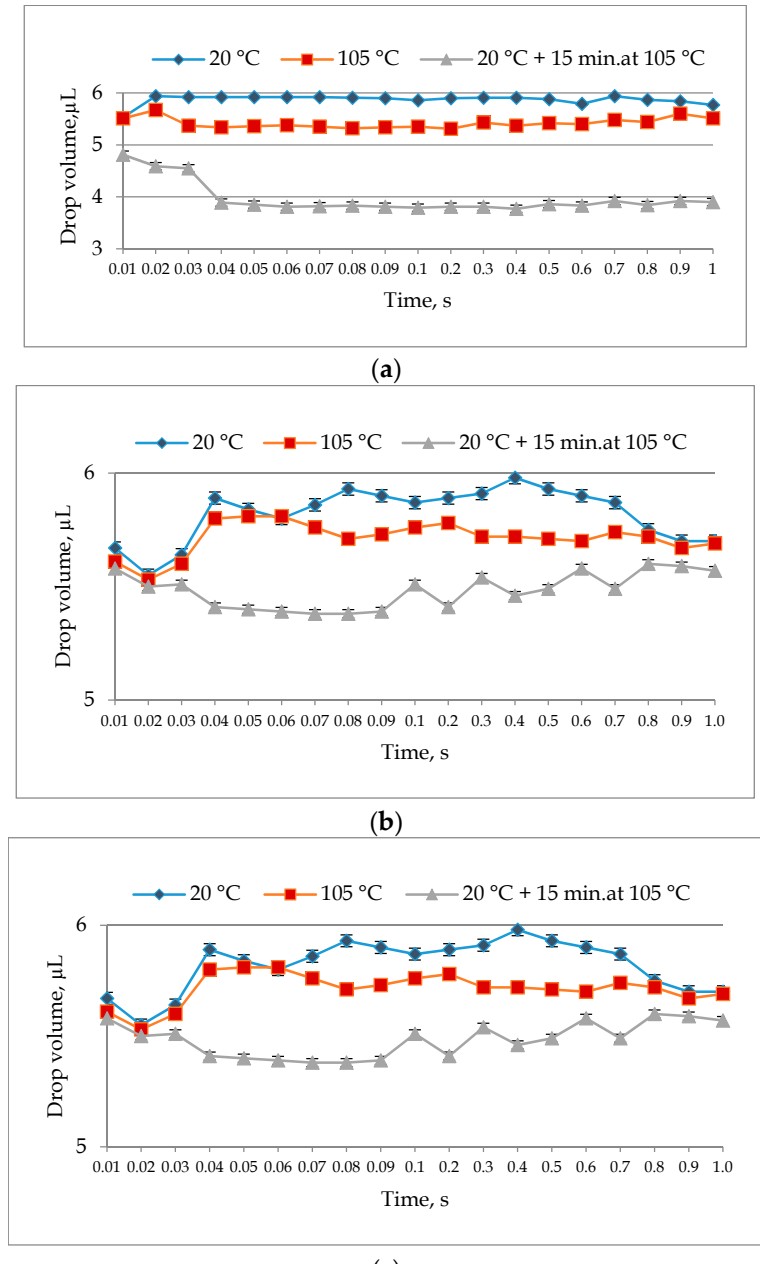

**Figure 11.** The evolution of the water absorption (drop base diameter) of coated samples in different drying conditions and latex contents: (**a**) 5 pph latex content; (**b**) 10 pph latex content; (**c**) 20 pph latex content.

## 4. Conclusions

In this paper, investigations concerning the influence of drying parameters on the structure and properties of paper coatings have been completed. Based on the obtained results, a correlation between the latex content in the coating formulation and the drying parameters of the coated papers has been found.

The drying conditions act on the structure and coating layer properties as a consequence of the fluid phase migration from the coating into base paper, which is higher at a low solids content in the coating formulation and a high drying temperature.

The increase in the drying temperature and latex content in the coating formulation has a synergic effect on reducing the coating layer void fraction and surface gloss.

The obtained results show that, at a high latex content in the coating formulation and a high drying temperature, a coating layer structure is obtained with large pores that allows light scattering and water absorption.

**Funding:** This research received no external funding.

**Institutional Review Board Statement:** Not applicable.

**Informed Consent Statement:** Not applicable.

**Data Availability Statement:** Data is contained within the article.

**Acknowledgments:** The author thank for support of the Research Centre for Environmental and Agriculture "Lunca", within Engineering and Agronomy Faculty in Brăila, "Dunărea de Jos" University of Galați, Romania.

**Conflicts of Interest:** The author declares no conflict of interest.

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
