# Peer review of "The Influence of Drying Conditions of Clay-Based Polymer Coatings on Coated Paper Properties"

_coatings, doi:10.3390/coatings11010012_

Round 1

Reviewer 1 Report

1. The manuscript entitled „ The influence of drying conditions on the structure and properties of paper coatings” aims to demonstrate the influence of water – based coating composition and the drying conditions on its surface properties.

The English in the manuscript should be improved before publishing. Due to wrong grammatical word ordering or tenses, comas missing in long sentences, some parts and sentences of the manuscript are confusing, and hard to follow. In addition, wrong word selection additionally causes some problems in clarity. I have pointed to some grammatical errors, not to all.

I am missing some scientific contribution of this study, the aim and the novelty. Too much attention is given to “coating colour drying” even though the results are more influenced by the coating composition. Namely, the opacity is known to be influenced by the pigment content. Some important facts should be revised and explained before publishing.   

From my opinion, the cited references are rather “old”. What is the reason for the fact that there are very few sources in the list of references over the past 5 years? (4 in total). One gets the feeling that either the topic is not relevant now or the author do not follow new research in this area.

For the improvement, please check the following comments:

2. Title: “The influence of drying conditions on the structure and properties of paper coatings” should be more specified. In the present form, I could point to paper coatings in general, but here it should be more specific, which coatings, based on what? I think that the author should mention the composition of the used coatings.

3. Abstract:

Line 9: please specify the composition of prepared coating. In the abstract, you refer to a water-born coating, and after that, you do not use that term anymore.

Line 12 (line 102): In our investigations? I think that one author cannot be in plural and talk about “our” investigation. Sounds better if “in this study the influence of drying conditions…” is used.

Lines 15-16: There are no “different levels of temperature and conditions”.  Please correct this.

Lines 17 – 18. “… for the samples of coated paper.” This part of sentence is not necessary since everything is said in the first part of the sentence.

4. Introduction:

Lines 23-28: The sentences should be checked in the terms of English. For example:

Generally, a coated paper is a layered composite, consisted of paper base covered on one side or both sides with a porous clay-based polymer coatings.

Line 24: These coatings..? Which coatings? Maybe “The coatings”.

Line 26: “Drying of paper coated…” – maybe “ drying of paper coating…”?

5. Line 35 (and through the whole manuscript): What is coating colour? Have you used coloured coating? Please explain. If not, you cannot talk about coating colour, only about coating or coating layer, coating formulations.

Lines 37-38: “However, in the coating layer during drying starts to develop the stress phenomenon.” What starts to develop stress phenomena? Water or coating. Please, check the English and ordering of words in the sentence. 

Line 39, 78: surface menisci… Do you mean surface tension?

Line 45: Please add the abbreviation of glass transition temperature into the brackets since you mention if after words as abbreviation.

6. Lines 60 – 62: Please check the word selection in the sentence. There are some problems in clarity.

7. Lines 74, 87 and 100: Figures 1-3. Are the Figures formed by the author or taken form some other references?

Line 74: Figure 1 is confusing. The term “paper” is formed coated paper or base paper? How the coating components can be from the downside of the paper? Paper, as a sheet, should than be presented as two black lines, between which pigments and binder are placed, or am I missing a point here?

8. Line 90: Paper does not have a homogeny surface, just in contrary; Its heterogenic surface plays an important role in the paper and ink interactions. 

Line 91: The binder has the supreme impact on printability [16-18]. Coating binder or printing ink binder, please be specific.

Lines 98-99: please take in consideration the word ordering: not “paper coated printability” but “coated paper printability”. Please correct it in the whole manuscript since it is repeated multiple times.

Materials and Methods

9. Line 114: have you used just one binder or several binders? I see only one mentioned here.

10. Lines 117-121: The coating formulations were prepared as only one parameter was varied through experiment (latex). When varying the latex content, you haven’t changed any other parameter, such as pigment content?

11. Line 127: The thickness of the coating layer is controlled by the diameter of the wire. It was wet coating deposition. Please specify which one, which rod value vas taken? What about the thickness of the dried coating?

12. Line 152- 153: is this a standard method? If yes, please specify which.

13. Line 155: µL not µl

Line 161: depth of 1-10 µ? Missing unit here.

14. Lines 154 – 159: how many measurements of water droplets were performed for each sample? Why haven’t you tested the base paper as well, in order to eliminate its influence on the measured parameters? The surface properties of base paper can explain the characteristics of coating, as well as its interaction with prepared coating.  

15. Lines 163-164: standard method of gloss determination is missing here

Line 190: please revise sentence as “… are presented in the Figures 6 and 7...”

16. Line 208: what about the SEM micrographs of 5 and 10 pph latex content?

17. Line 211: please revise as “… coated paper samples…”

Line 214: You have only tested the penetration of the water on coated paper, not several liquids. Please revise.

Lines 215-217: “This parameter provides information about the capacity of fluids to adhere and wetting the surface of certain substrates, as well as the useful information regarding the substrate capacity to absorb the printing inks.” -  This is correct but only in the case if the penetration was evaluated by different test liquids, with different surface tensions. In this case, where only water was used, this should be taken with caution since printing inks can vary in their composition, they are not only water based. The water contact angle in his case can only provide an information about the hydrophobicity of the coated paper surface, and the influence of the coating composition on the surface characteristics. It cannot be used for the prediction of printing ink absorption into the paper.

Lines 220 -221: Figure 9. There is missing the standard deviation for the green line.

18. Lines 230-231: “By latex migration into base paper the hydrophobicity level and contact angle value of coating layer are reduced.” What is the surface in general due to obtained contact angles? What about the minimum measured value for contact angle in all cases?

19. Lines 239-247: how was the base diameter measured?

20. Lines 245 – 247: increase of the drop base is only due to saturation or something else maybe? Surface free energy?  

Author Response

Dear reviewer,

Thank you very much for your useful observations, comments and recommendations. Please, find bellow, point by point, the response at your comments:

Reviewer comments

1. The manuscript entitled „ The influence of drying conditions on the structure and properties of paper coatings” aims to demonstrate the influence of water – based coating composition and the drying conditions on its surface properties.

The English in the manuscript should be improved before publishing. Due to wrong grammatical word ordering or tenses, comas missing in long sentences, some parts and sentences of the manuscript are confusing, and hard to follow. In addition, wrong word selection additionally causes some problems in clarity. I have pointed to some grammatical errors, not to all.

I am missing some scientific contribution of this study, the aim and the novelty. Too much attention is given to “coating colour drying” even though the results are more influenced by the coating composition. Namely, the opacity is known to be influenced by the pigment content. Some important facts should be revised and explained before publishing.   

From my opinion, the cited references are rather “old”. What is the reason for the fact that there are very few sources in the list of references over the past 5 years? (4 in total). One gets the feeling that either the topic is not relevant now or the author do not follow new research in this area.

If the manuscript will be accepted we will send it to MDPI English editing service. Our staff representatives approves the publishing charge (including English service)  only after the manuscript is accepted for publishing.

The paper coating is a research area which attracted attention in the last times as environmentally friendly alternatives for packaging applications. From this perspective, paper appears as ideal packaging material, having some advantages as high recyclability, biodegradability and compostability, is from renewable raw material, comparing with plastics products. To improve the paper properties, coatings are applied on their surface  and all the influences must be analysed (the coating composition, drying or finishing parameters etc.) to obtain a proper product which can be used as packaging. In this context was performed the present study.

In the revised manuscript and in the  references section, new references were added .

2. Title: 

Reviewer comments

 “The influence of drying conditions on the structure and properties of paper coatings” should be more specified. In the present form, I could point to paper coatings in general, but here it should be more specific, which coatings, based on what? I think that the author should mention the composition of the used coatings.

The new title: “The influence of drying conditions of the clay- based polymer coatings on the coated paper properties”

3. Abstract:

Reviewer comments

Line 9: please specify the composition of prepared coating. In the abstract, you refer to a water-born coating, and after that, you do not use that term anymore.

Line 12 (line 102): In our investigations? I think that one author cannot be in plural and talk about “our” investigation. Sounds better if “in this study the influence of drying conditions…” is used.

Lines 15-16: There are no “different levels of temperature and conditions”.  Please correct this.

Lines 17 – 18. “… for the samples of coated paper.” This part of sentence is not necessary since everything is said in the first part of the sentence.

In the revised  manuscript whole abstract section was revised.

4. Introduction:

Reviewer comments

Lines 23-28: The sentences should be checked in the terms of English. For example:

Generally, a coated paper is a layered composite, consisted of paper base covered on one side or both sides with a porous clay-based polymer coatings.

Line 24: These coatings..? Which coatings? Maybe “The coatings”.

Line 26: “Drying of paper coated…” – maybe “ drying of paper coating…”?

All of these sentences were revised .

Reviewer comments

5. Line 35 (and through the whole manuscript): What is coating colour? Have you used coloured coating? Please explain. If not, you cannot talk about coating colour, only about coating or coating layer, coating formulations.

The term of coating colour is usually used in coatings. This explains the mixture of all coatings components and is considered as paste form. Coating formulations is used rather for coatings recipe. Of course, I don’t use the coloured coating. For more clarity,  in the revised manuscript  I removed this term and the term of coating formulation was introduced.

6. Reviewer comments

Lines 37-38: “However, in the coating layer during drying starts to develop the stress phenomenon.” What starts to develop stress phenomena? Water or coating. Please, check the English and ordering of words in the sentence. 

Line 39, 78: surface menisci… Do you mean surface tension?

Line 45: Please add the abbreviation of glass transition temperature into the brackets since you mention if after words as abbreviation.

All of these sentences were revised .

7. Reviewer comments

Lines 74, 87 and 100: Figures 1-3. Are the Figures formed by the author or taken form some other references?

 Line 74: Figure 1 is confusing. The term “paper” is formed coated paper or base paper? How the coating components can be from the downside of the paper? Paper, as a sheet, should than be presented as two black lines, between which pigments and binder are placed, or am I missing a point here?

These figures are formed by the author. Indeed in the figure 1 was a regrettable mistake which was corrected in the revised  manuscript.

8. Reviewer comments

Line 90: Paper does not have a homogeny surface, just in contrary; Its heterogenic surface plays an important role in the paper and ink interactions. 

Line 91: The binder has the supreme impact on printability [16-18]. Coating binder or printing ink binder, please be specific.

Lines 98-99: please take in consideration the word ordering: not “paper coated printability” but “coated paper printability”. Please correct it in the whole manuscript since it is repeated multiple times.

All of these sentences were revised .

 Materials and Methods

9. Reviewer comments

Line 114: have you used just one binder or several binders? I see only one mentioned here.

As is presented in the table 2, only one type of binder was used in experiments.

10. Reviewer comments

Lines 117-121: The coating formulations were prepared as only one parameter was varied through experiment (latex). When varying the latex content, you haven’t changed any other parameter, such as pigment content?

In the coating formulations only latex content was varied. All the components was calculated and added as parts per hundred parts of dried weight pigment (pph).

11. Reviewer comments

Line 127: The thickness of the coating layer is controlled by the diameter of the wire. It was wet coating deposition. Please specify which one, which rod value vas taken? What about the thickness of the dried coating?

A rod 3 value was used that corresponds to a dry coating layer thickness of about 10 μm.

12. Reviewer comments

Line 152- 153: is this a standard method? If yes, please specify which.

The void fraction is not a standard method. In the manuscript are indicated the refences for this

13. Reviewer comments

Line 155: µL not µl

Line 161: depth of 1-10 µ? Missing unit here.

All of these sentences were revised .

14. Reviewer comments

Lines 154 – 159: how many measurements of water droplets were performed for each sample? Why haven’t you tested the base paper as well, in order to eliminate its influence on the measured parameters? The surface properties of base paper can explain the characteristics of coating, as well as its interaction with prepared coating.

The 3 measurements of water droplets were performed for each sample .  The comment is correct that the base paper properties influence the coatings and coated paper properties. In the experiment  a sized base paper (with Cobb value about 22.6 g/m2) was used and in this respect the contact angle was not measured.

 15. Reviewer comments

Lines 163-164: standard method of gloss determination is missing here

Line 190: please revise sentence as “… are presented in the Figures 6 and 7...”

All of these sentences were revised  and standard method for gloss was introduced.

16. Reviewer comments

Line 208: what about the SEM micrographs of 5 and 10 pph latex content?

The SEM micrographs were performed only samples with 20 pph latex content, to emphasize the effect of high latex content in coatings. Between 5 and 10 pph there are no significant modification in the coating layer structure indifferent of drying conditions.

17. Reviewer comments

Line 211: please revise as “… coated paper samples…”

Line 214: You have only tested the penetration of the water on coated paper, not several liquids. Please revise.

Lines 215-217: “This parameter provides information about the capacity of fluids to adhere and wetting the surface of certain substrates, as well as the useful information regarding the substrate capacity to absorb the printing inks.” -  This is correct but only in the case if the penetration was evaluated by different test liquids, with different surface tensions. In this case, where only water was used, this should be taken with caution since printing inks can vary in their composition, they are not only water based. The water contact angle in his case can only provide an information about the hydrophobicity of the coated paper surface, and the influence of the coating composition on the surface characteristics. It cannot be used for the prediction of printing ink absorption into the paper.

Lines 220 -221: Figure 9. There is missing the standard deviation for the green line.

All of these sentences  and Figure 9 were revised. The standard deviation is for all lines. It can be seen at large zoom of figure.

18. Reviewer comments

Lines 230-231: “By latex migration into base paper the hydrophobicity level and contact angle value of coating layer are reduced.” What is the surface in general due to obtained contact angles? What about the minimum measured value for contact angle in all cases?

A low level of coated surface hydrophobicity is obtained for all the coating formulations. Based on the other research in this fields, I consider that the introducing in the coating formulations a water retention additive (i.e. carboxymethyl cellulose) and increasing of its solids content (to 60%) seems to be efficient routes to reduce of coating dewatering rate and obtaining the coated surfaces with uniform characteristics and appropriate hydrophobicity.

19. Reviewer comments

Lines 239-247: how was the base diameter measured?

The base diameter of droplet was authomatically registered when the contact angle was measured  with  Fibrodat device.

20. Reviewer comments

Lines 245 – 247: increase of the drop base is only due to saturation or something else maybe? Surface free energy?  

Response of author

Yes , the surface free energy influences the contact angle. When the paper surface free energy increases, a low level of contact angle is obtained

Reviewer 2 Report

1. The paper is potentially interesting, but the experimental planning is quite poor since very few conditions were tested. Moreover, it is not clear why no more temperature range where tested as well as how the 15 minutes at 105ºC was chosen.  

2. How much time the U2 drying condition at 105ºC was applied?

3. All the figures should report the gloss measure of the control without latex using the same temperatures.

4. Photographs of the contact angle tests are welcomes. Moreover, the results of Figure 9 and 10 should be showed for more time (at least 5 s)

5. Optical micrograph photos of the coated papers showing the cracks should be provided.

Author Response

Dear reviewer,

Thank you very much for your useful observations, comments and recommendations. Please, find bellow the response of your observations:

1. Reviewer comments

The paper is potentially interesting, but the experimental planning is quite poor since very few conditions were tested. Moreover, it is not clear why no more temperature range where tested as well as how the 15 minutes at 105ºC was chosen.  

In the industrial plant the coated papers are dried at 105-106°C and lab samples at room temperature. In this respect, it was used only two values of drying temperature. The time of 15 min was chosen  taking into account the coating layer grammage (15-16 g/m2) and coating color solids content that was relatively low (about 45 %).

2. Reviewer comments

How much time the U2 drying condition at 105ºC was applied?

The drying time of U2 samples was 45 minutes. It was mentioned in revised  manuscript (with red color)

3. Reviewer comments

All the figures should report the gloss measure of the control without latex using the same temperatures.

Based on the fact that these laboratory coated paper samples were not calendered (when the gloss of coated surface is intensified and more influenced by latex content) I considered  only the experiments with latex.

4. Reviewer comments

Photographs of the contact angle tests are welcomes. Moreover, the results of Figure 9 and 10 should be showed for more time (at least 5 s)

As is observed from figures 9 (10 after revision) and 10 (11 after revision), after 0.3 s, the surface of coated paper became saturated and the rate of change of both contact angle and drop base diameter is low. This is the reason that I considered the contact time of 1 s.

5. Reviewer comments

Optical micrograph photos of the coated papers showing the cracks should be provided.

The cracks in coating structure is marked with red line onto SEM micrographs  (figure  9) in the revised manuscript.

Round 2

Reviewer 1 Report

The author respond to all my comments. Only the English language should be checked before publishing.

Reviewer 2 Report

The authors' answers and manuscript modification are satisfactory. I recommend it for publication.